# PFMBench: Protein Foundation Model Benchmark

## Abstract

This study investigates the current landscape and future directions of protein foundation model research. While recent advancements have transformed protein science and engineering, the field lacks a comprehensive benchmark for fair evaluation and in-depth understanding. Since ESM-1B, numerous protein foundation models have emerged, each with unique datasets and methodologies. However, evaluations often focus on limited tasks tailored to specific models, hindering insights into broader generalization and limitations. Specifically, researchers struggle to understand the relationships between tasks, assess how well current models perform across them, and determine the criteria in developing new foundation models. To fill this gap, we present PFMBench, a comprehensive benchmark evaluating protein foundation models across 38 tasks spanning 8 key areas of protein science. Through hundreds of experiments on 17 state-of-the-art models across 38 tasks, PFMBench reveals the inherent correlations between tasks, identifies top-performing models, and provides a streamlined evaluation protocol. Code will be released upon acceptance.

## 1 Introduction

Protein foundation models (PFMs) have garnered significant attention in recent years for their transformative potential in protein science and engineering. By training on large-scale protein datasets, these models capture intricate relationships between sequences, structures, and functions. Since the debut of ESM-1B Rives et al. (2021) in 2021, a diverse array of PFMs—spanning various architectures and training paradigms—has emerged Rives et al. (2021); Lin et al. (2023); Hayes et al. (2025); Elnaggar et al. (2021); Madani et al. (2023); Ferruz et al. (2022); Tan et al. (2025); Zhou et al. (2025); Elnaggar et al. (2023); Chen et al. (2024); Wang et al. (2024b); Su et al.; 2024); Xu et al. (2023); Bjerregaard et al. (2025); Guo et al. (2025); Li et al. (2024). Despite this rapid progress, prior models like ESM2 Lin et al. (2023) still dominate many bioengineering applications. This raises several pressing questions: Has the field reached a plateau and what is the next frontier for PFMs? Thus, a comprehensive and systematic benchmark is urgently needed.

Comparison of PFMBench with existing benchmarks. #PFM: number of PFMs (>500M), MM: multimodal PFMs, #Tasks: task count, Protocol: simplified evaluation protocol.

| | | #PFM | MM | #Tasks | Protocol |
|---|---|---|---|---|---|
| TAPE | NeurIPS 2019 | 0 | ✗ | 4 | ✗ |
| PEER | NeurIPS 2022 | 1 | ✗ | 14 | ✗ |
| CARE | NeurIPS 2024 | 0 | ✗ | 2 | ✗ |
| Venus | Arxiv 2025 | 3 | ✗ | 22 | ✗ |
| Our | | 14 | ✓ | 38 | ✓ |

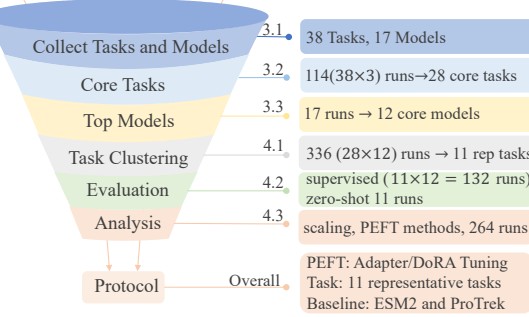

Figure 1: PFMBench: More tasks, multimodal PFMs, a simplified protocol, and hierarchical analysis.

Previous benchmarking efforts for protein models have either covered a limited set of tasks or were not explicitly designed for evaluating foundation models, as shown in Fig. 1. In the context of protein foundation models (PFMs)—typically defined as models with at least 500 million parameters—most existing benchmarks fall short of providing comprehensive evaluation. For example, TAPE Rao et al. (2019) assessed architectures such as Transformers Vaswani et al. (2017), LSTMs Hochreiter & Schmidhuber (1997), and ResNets He et al. (2016) across four tasks, but did not include any large-scale PFMs. PEER Xu et al. (2022) evaluated models on 14 tasks but was limited to sequence-based

architectures, with only ESM-1B Rives et al. (2021) exceeding the 500-million-parameter threshold. CARE Yang et al. (2024) focused narrowly on two enzyme-related tasks: classification and retrieval. More recently, VenusFactory Tan et al. (2025) introduced a unified benchmark spanning 22 tasks across five functional categories. However, it reported results for only three large sequence-based models, such as ESM2 Lin et al. (2023), Ankh Elnaggar et al. (2023), and ProtT5 Elnaggar et al. (2021), limiting its ability to capture the full spectrum of modern PFMs.

Multimodal PFMs are understudied in existing benchmarks, despite the field's rapid shift toward models that integrate sequence, structure, and functional data. For example, ESM3 Hayes et al. (2025), GearNet Zhang et al. (b), and SaProt Su et al. have demonstrated strong performance on specialized tasks such as protein design and function prediction. However, their evaluations are often limited in scope, focusing on specific tasks or datasets, which impedes a systematic understanding of their limitations, generalizability, and cross-task performance. For instance, while ESM3 excels in protein design, its ability to generalize to other tasks remains largely unexplored. Similarly, GearNet and SaProt have shown promise in certain tasks, but their performance across broader protein function landscapes has yet to be thoroughly assessed. Consequently, it remains unclear under what conditions and how multimodal PFMs contribute to improved generalization capabilities.

A benchmark should not merely serve as a collection of tasks and models—it should also provide a streamlined protocol for model development. As both tasks and models become increasingly complex, exhaustively evaluating all models across all tasks becomes impractical and often fails to yield actionable insights. A more effective approach is to uncover the underlying relationships between tasks, identify a representative subset of tasks, and select a diverse yet informative set of models for focused evaluation. This strategy enables the benchmark to help researchers identify top-performing models for specific tasks and guide the development of new models—serving as a blueprint for future model evaluation, selection, and design.

To address this gap, we introduce PFMBench—a unified and comprehensive benchmark suite for protein foundation models. PFMBench spans 38 tasks across 8 categories, encompassing 19 sequence-based, sequence-structure, sequence-function, and multimodal PFMs. Both datasets and models are carefully curated to ensure robust, fair and meaningful comparisons. Through extensive evaluation, PFMBench offers detailed insights into the strengths and limitations of modern PFMs, and provide a simplified and useful protocol for future PFM development.

## 2 RELATED WORK

**Protein Foundation Models.** Protein foundation models (PFMs) have witnessed exponential growth in recent years, revolutionizing computational biology through self-supervised learning on vast protein sequence datasets. ESM-1b Rives et al. (2021) pioneered large-scale protein modeling with a 650M parameter transformer trained on 65 million protein sequences via masked language modeling. This trajectory continued with ESM-2 and ESMC models Lin et al. (2023), which demonstrated enhanced representation learning for protein structure and function through refined architecture and expanded training data. The ESM family evolved further with ESM3 Hayes et al. (2025), scaling to 98B parameters and incorporating structure-aware training to achieve state-of-the-art performance on zero-shot fitness prediction and structure modeling. ProtT5 Elnaggar et al. (2021) adapted the T5 architecture to proteins, scaling to 3B and 11B parameters with span-masking objectives, establishing strong baselines for protein sequence-to-sequence tasks. The generative approach was pioneered by ProGen Madani et al. (2023), a 1.2B parameter conditional generation model, and ProtGPT2 Ferruz et al. (2022), a 738M parameter GPT-2-based model for de novo protein sequence generation. VenusPLM Tan et al. (2025) employed transformer-based architectures with modular fine-tuning capabilities for enzyme engineering and protein function prediction. Multimodal approaches emerged with ProtCLIP Zhou et al. (2025), aligning protein sequences with biological text through function-informed pre-training. ANKH Elnaggar et al. (2023) built upon ProtT5's architecture to optimize data efficiency through systematic ablation studies. xTrimoPGLM Chen et al. (2024) adopt GLM's training paradigm to protein sequences, expanding the model size to 100B. Other significant contributions include DPLM Wang et al. (2024b), leveraging deep learning for protein language modeling; SaProt Su et al., focusing on structure-aware protein representation learning; ProtRek Su et al. (2024), specialized in protein sequence retrieval and knowledge integration; and ProST Xu et al. (2023), which incorporates biomedical texts to guide protein function learning. Together, these

diverse foundation models have transformed protein research by enabling unprecedented advances in structure prediction, functional annotation, and protein design through their ability to learn complex evolutionary and structural patterns from sequence data.

**Protein Benchmarks.** Protein foundation model benchmarks have evolved significantly, transitioning from early efforts like TAPE Rao et al. (2019), which evaluated small models on a limited set of tasks, to more comprehensive frameworks. PEER Xu et al. (2022) expanded the scope by introducing a multi-task benchmark encompassing diverse protein understanding tasks, including function prediction and protein-protein interactions. BeProf Wang et al. (2024a) further contributed by evaluating deep learning-based protein function prediction models in different application scenarios. Recent benchmarks like VenusFactory Tan et al. (2025) have integrated a broader range of pre-trained models and datasets, yet they often lack consideration for multimodal approaches. Beyond predictive benchmarks, initiatives like ProteinGym Notin et al. (2023), ProteinInvBench Gao et al. (2023) and ProteinBench Ye et al. (2024) have introduced frameworks for evaluating protein mutation effects, inverse folding and protein design, respectively. These benchmarks have progressively incorporated more diverse tasks, models—including large pre-trained language models and multimodal approaches—and sophisticated evaluation metrics, thereby playing a crucial role in tracking progress, identifying state-of-the-art methods, and guiding future research. However, current benchmarks do not foucus on protein foundation models, especially multimodal foundation models, also do not provide a streamlined evaluation protocol for these models.

**Parameter-Efficient Fine-Tuning.** Recent advances in parameter-efficient fine-tuning (PEFT) have enabled the adaptation of large pre-trained models by updating only a small subset of their parameters. Adapter-based methods insert trainable modules between frozen layers Houlsby et al. (2019); Pfeiffer et al. (2020), while Low-Rank Adaptation (LoRA) approximates weight updates using low-rank matrices Hu et al. (2022a). Prompt-based techniques—such as prefix tuning Li & Liang (2021) and prompt tuning Lester et al. (2021)—optimize soft prompts within the input embeddings, avoiding changes to the model weights. Other approaches, including BitFit (which updates only bias terms) Zaken et al. (2022), IA3 (which scales intermediate activations) Liu et al. (2022), and QLoRA (which enables quantized fine-tuning) Dettmers et al. (2023), further improve efficiency. Hybrid strategies that combine multiple techniques have also emerged He et al.. Recent innovations include AdaLoRA, which dynamically adjusts rank allocation during training Zhang et al. (2023); MoeLoRA, which integrates mixture-of-experts into LoRA for enhanced scalability Wu et al. (2024); DoRA, which decomposes weights into magnitude and direction for targeted adaptation Mao et al. (2024); and LoCA, which introduces location-aware cosine adaptation for more precise updates Du et al. (2025). Collectively, these developments continue to improve the efficiency, flexibility, and effectiveness of PEFT for large language models. This research select Adapter, LoRA, AdaLoRA, DoRA and IA3 as the representative methods for performance comparison.

# 3 METHOD

## 3.1 PFMBENCH FRAMEWORK

**Framework.** As shown in Figure 2, PFMBench comprises three main components: (1) a user-friendly interface, (2) a suite of downstream tasks, and (3) a comprehensive collection of foundation models. Designed with modularity in mind, the framework allows users to swap components and customize the evaluation process with ease. We employ Hydra to parse configuration files and PyTorch Lightning to manage model fine-tuning. To our knowledge, PFMBench is the largest and most comprehensive benchmark for protein foundation models, covering 38 tasks across 17 models.

**Data Contribution.** For each dataset, we retrieve protein structures from the AF2DB Varadi et al. (2022) when available; otherwise, we use ESMFold Lin et al. (2023) to generate the rank-1 protein structure. To standardize evaluation, we enforce a 30% sequence similarity cutoff when splitting data, resulting in an 8:1:1 ratio for training, validation, and test sets. Mutation datasets are exempt from this splitting due to their high similarity to wild-type sequences; thus, we retain their original train/validation/test partitions.

**Protocol Contribution.** Evaluating all models and tasks is impractical, especially when aiming to provide guidance for developing new foundation models. We believe that simplifying the selection of tasks and models is equally important, as it highlights the key insights. Through hundreds of experiments, we provide a hierarchical analysis that results in a streamlined protocol: (1) Baseline: select either the sequence-only ESM2 or the multimodal ProTrek; (2) Task: filter 11 representative tasks from the original 38 tasks; (3) PEFT: adopt either the transformer-adapter or the DoRA tuning.

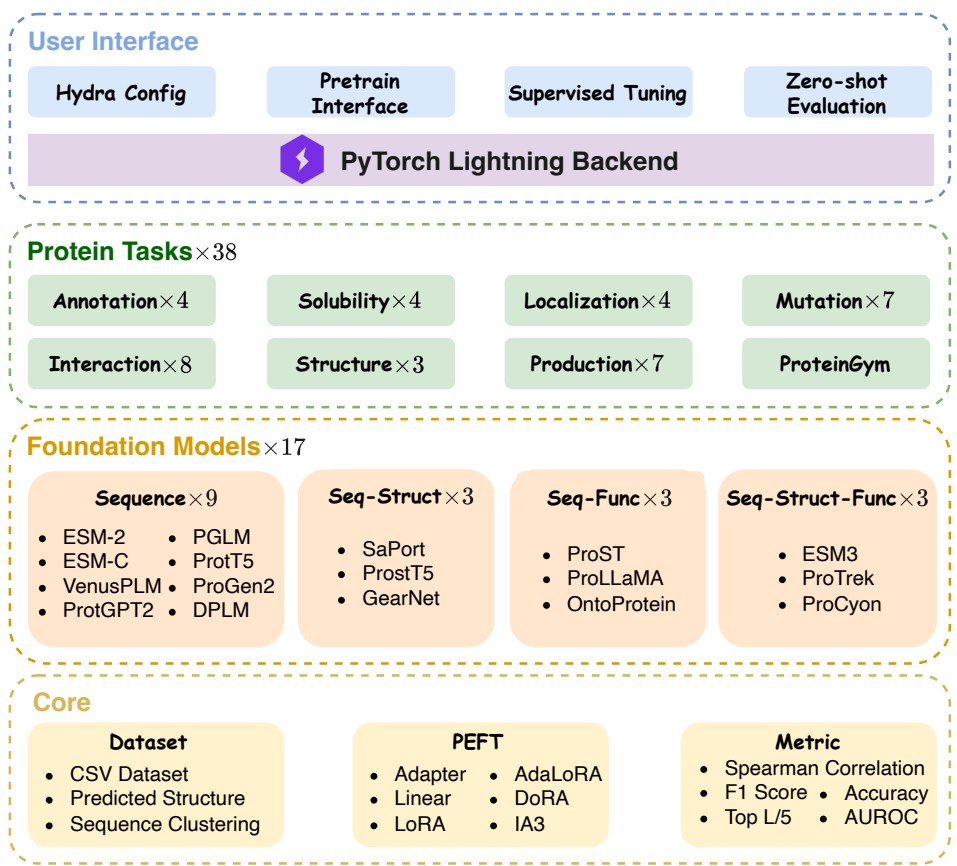

Figure 2: The Overall framework of PFMBench. The framework includes: (1) a user-friendly interface, (2) enumerious downstream tasks, and (3) a comprehensive set of foundation models. Diverse datasets, parameter-efficient tuning methods, and evaluation metrics are integrated. The modular design allows users to easily swap components, customize models, tasks and metrics.

## 3.2 SUPPORTED TASKS

**Core Tasks.** PFMBench includes 38 tasks spanning diverse domains, covering both supervised and zero-shot learning. Supervised tasks are grouped into seven categories: Annotation, Solubility, Localization, Mutation, Interaction, Structure, and Production. Definitions, metrics, and impacts for each category are detailed in Appendix A.1. Datasets are split into training, validation, and test sets using an 8:1:1 ratio with a 30% sequence similarity threshold, except for mutation datasets. We evaluate ESM2-Adapter on all tasks, averaging results over three runs, with bias calculated as the absolute difference between the best and worst runs divided by the average performance (see Table 1). To ensure unbiased evaluation, we designates 28 tasks with a bias below 5% as core tasks.

## 3.3 SUPPORTED MODELS

**Core Models.** PFMBench supports a broad spectrum of protein foundation models, as summarized in Table 2. To ensure a fair comparison, we select models with parameter counts close to 1B when multiple versions are available. Based on input data modalities, the models are categorized into four

Table 1: PFMBench Tasks span eight categories, detailing training, validation, and test sample counts per task with references. Symbols △ and ☆ indicate datasets with sequence or sequence-structure pairs, as used in benchmarks like TAPE Rao et al. (2019), PEER Xu et al. (2022), Venus Tan et al. (2025), and our framework. ESM2-Adapter's mean and bias performance are shown, with core tasks having bias below 5%.

| Task | Metric | Train | Val | Test | TAPE | Peer | Venus | Our | Mean | Bias(%) | Core |
|---|---|---|---|---|---|---|---|---|---|---|---|
| **Annotation** | | | | | | | | | | | |
| Cellular Component Ashburner et al. (2000) | F1 Score | 11196 | 1398 | 1400 | | | ☆ | ☆ | 0.6130 | 0.26% | ✓ |
| Molecular Function Ashburner et al. (2000) | F1 Score | 22291 | 2785 | 2787 | | | ☆ | ☆ | 0.6488 | 0.38% | ✓ |
| Biological Process Ashburner et al. (2000) | F1 Score | 21395 | 2662 | 2664 | | | ☆ | ☆ | 0.5412 | 0.79% | ✓ |
| Enzyme Commission Bairoch (2000) | F1 Score | 13090 | 1465 | 1604 | | | ☆ | ☆ | 0.7379 | 0.09% | ✓ |
| **Solubility** | | | | | | | | | | | |
| DeepSol Khurana et al. (2018) | AUROC | 55465 | 6932 | 6934 | | △ | ☆ | ☆ | 0.8467 | 0.23% | ✓ |
| DeepSoluE Wang & Zou (2023) | AUROC | 11627 | 1452 | 1454 | | | ☆ | ☆ | 0.7699 | 1.10% | ✓ |
| ProtSolM Tan et al. (2024b) | AUROC | 57378 | 7171 | 7173 | | | ☆ | ☆ | 0.8572 | 0.93% | ✓ |
| eSOL Chen et al. (2021) | Spearman | 2481 | 309 | 311 | | | ☆ | ☆ | 0.2761 | 38.3% | |
| **Localization** | | | | | | | | | | | |
| DeepLoc Multi Almagro Armenteros et al. (2017) | Accuracy | 6992 | 749 | 751 | | △ | ☆ | ☆ | 0.7666 | 1.27% | ✓ |
| DeepLoc2 Multi Thumuluri et al. (2022) | F1 Score | 21949 | 2743 | 2744 | | △ | ☆ | ☆ | 0.7505 | 0.16% | ✓ |
| DeepLoc Binary Almagro Armenteros et al. (2017) | AUROC | 6887 | 846 | 848 | | | | ☆ | 0.9338 | 0.42% | ✓ |
| Sorting Signal Thumuluri et al. (2022) | F1 Score | 1484 | 185 | 186 | | △ | | ☆ | 0.8598 | 0.24% | ✓ |
| **Mutation** | | | | | | | | | | | |
| PETA_CHS_Sol Tan et al. (2024a) | Spearman | 3872 | 484 | 484 | | | △ | ☆ | 0.2738 | 12.5% | |
| PETA_LGK_Sol Tan et al. (2024a) | Spearman | 15308 | 1914 | 1914 | | | △ | ☆ | 0.1558 | 21.7% | |
| PETA_TEM_Sol Tan et al. (2024a) | Spearman | 6444 | 808 | 808 | | | △ | ☆ | 0.1433 | 27.0% | |
| FLIP_AAV Dallago et al. | Spearman | 66066 | 16517 | 16517 | | | △ | ☆ | 0.9412 | 0.13% | ✓ |
| FLIP_GB1 Dallago et al. | Spearman | 6988 | 1745 | 1745 | | | △ | ☆ | 0.9517 | 0.13% | ✓ |
| TAPE_Stability Rao et al. (2019) | Spearman | 55182 | 6897 | 6898 | △ | △ | △ | ☆ | 0.3211 | 4.01% | ✓ |
| TAPE_Fluorescence Rao et al. (2019) | Spearman | 21446 | 5362 | 27217 | △ | △ | △ | ☆ | 0.6812 | 0.21% | ✓ |
| β-lactamase activity Gray et al. (2018) | Spearman | 4158 | 520 | 520 | | △ | | ☆ | 0.5740 | 21.6% | |
| **Interaction** | | | | | | | | | | | |
| Human-PPI Pan et al. (2010) | AUROC | 30133 | 270 | 195 | | △ | | ☆ | 0.4828 | 0.00% | ✓ |
| Yeast-PPI Guo et al. (2008) | AUROC | 4157 | 83 | 335 | | △ | | ☆ | 0.5343 | 12.8% | |
| PPI affinity Moal & Fernández-Recio (2012) | Spearman | 2421 | 203 | 326 | | △ | | ☆ | -0.0047 | 114.3% | |
| PDBbind Liu et al. (2017) | Spearman | 14687 | 1835 | 1836 | | △ | | ☆ | 0.1677 | 4.14% | ✓ |
| BindingDB Liu et al. (2007) | Spearman | 9039 | 1115 | 1139 | | △ | | ☆ | 0.1922 | 3.02% | ✓ |
| Metal ion Binding Hu et al. (2022b) | Accuracy | 5740 | 717 | 718 | | | ☆ | ☆ | 0.7066 | 2.43% | ✓ |
| Pept.HLA/MHC Aff. Wu et al. (2023) | AUROC | 57357 | 7008 | 8406 | | | | ☆ | 0.9631 | 0.00% | ✓ |
| TCR PMHC Affinity Koyama et al. (2023) | AUROC | 19264 | 2265 | 2482 | | | | ☆ | 0.9312 | 0.00% | ✓ |
| **Structure** | | | | | | | | | | | |
| Contact prediction Yang et al. (2020) | Top L/5 | 12005 | 1500 | 1501 | △ | △ | | ☆ | 0.7199 | 0.40% | ✓ |
| Fold classification Lo Conte et al. (2000) | Accuracy | 13034 | 1628 | 1630 | | △ | | ☆ | 0.7859 | 0.31% | ✓ |
| Secondary structure Klausen et al. (2019) | Accuracy | 67007 | 8365 | 8262 | △ | △ | | ☆ | 0.7601 | 0.00% | ✓ |
| **Production** | | | | | | | | | | | |
| Optimal PH Gado et al. (2023) | Spearman | 7669 | 958 | 959 | | | | ☆ | 0.0564 | 17.6% | |
| DeepET_Topt Li et al. (2022b) | Spearman | 1479 | 184 | 185 | | | ☆ | ☆ | 0.2628 | 7.00% | |
| Cloning CLF Wang et al. (2014) | AUROC | 22223 | 2777 | 2778 | | | | ☆ | 0.8160 | 0.51% | ✓ |
| Material Production Wang et al. (2014) | Accuracy | 22196 | 2773 | 2775 | | | | ☆ | 0.7982 | 0.00% | ✓ |
| Enzyme Eff. Li et al. (2022a) | Spearman | 10363 | 1298 | 1290 | | | | ☆ | 0.2173 | 58.2% | |
| Antib. Res. Hu et al. (2022b) | Accuracy | 2703 | 336 | 339 | | | | ☆ | 0.6185 | 2.23% | ✓ |
| Thermostability Jarzab et al. (2020) | AUROC | 33474 | 4184 | 4184 | | | ☆ | ☆ | 0.9553 | 1.27% | ✓ |
| **Zero-shot** | | | | | | | | | | | |
| ProteinGym Notin et al. (2023) | Spearman | | | | | | ☆ | ☆ | 0.4390 | 0% | ✓ |

groups: (1) sequence-only models, (2) sequence-structure models, (3) sequence-function models, and (4) sequence-structure-function models. To establish a consistent evaluation baseline, we assess all models on the enzyme commission (EC) classification task under the adapter tuning setting. Models that achieve at least 85% of ESM2's performance are selected as core models for further evaluation. For detailed reasons regarding the adoption of EC as a selective task, please refer to Appendix A.5.

### 3.4 SUPPORTED TUNING METHODS

PFMBench offers diverse parameter efficient fine-tuning (PEFT) methods: linear probing, adapter tuning, $IA^3$, LoRA, AdaLoRA, and DoRA, with a unified interface for seamless switching.

**Adapter Tuning & Linear Probing.** We extract features using the pretrained model and employ a 6-layer transformer as a task-specific adapter with a hidden size of 480 and 20 attention heads. In Linear probing setting, we the transformer adapter is replaced with a linear layer. Without additional explanation, we report adapter tuning results in the main text.

**Other Tuning Methods.** **LoRA** decomposes attention and feedforward layer weight updates into the product of two low-rank matrices, which are the only trainable components during finetuning Hu et al. (2022a). **$IA^3$** introduces trainable multiplicative scalars into the attention and MLP sublayers, modulating the flow of information through each component Liu et al. (2022). **AdaLoRA** dynamically

Table 2: Models in PFMBench. The table lists the models, architecture types, number of parameters, publication states, code sources. We report the Enzyme Commission (EC) results.

| Model | Core | Architecture | # Params | Publication | EC | Code |
|---|---|---|---|---|---|---|
| **Sequence** | | | | | | |
| ESM-2 Lin et al. (2023) | ✓ | Encoder | 650M | Science 23 | 0.7358 | HF |
| VenusPLM Tan et al. (2025) | ✓ | Encoder | 300M | Arxiv 25 | 0.7519 | HF |
| ESM-C | ✓ | Encoder | 600M | Blog 25 | 0.7169 | HF |
| ProtGPT2 Ferruz et al. (2022) | ✓ | Decoder | 738M | Nat. Commun. 22 | 0.6969 | HF |
| ProGen2 Nijkamp et al. (2023) | | Decoder | 764M | Cell Syst. 23 | 0.6198 | GitHub |
| xTrimoPGLM Chen et al. (2024) | ✓ | Encoder-Decoder | 1B | Nat. Methods 25 | 0.7466 | HF |
| ProtT5 Elnaggar et al. (2021) | ✓ | Encoder-Decoder | 3B | TPAMI 21 | 0.7620 | HF |
| DPLM Wang et al. (2024b) | ✓ | Encoder+Diffusion | 650M | ICLM 24 | 0.7552 | GitHub |
| **Sequence-Structure** | | | | | | |
| SaPort Su et al. | ✓ | Encoder | 650M | ICLR 24 | 0.7514 | HF |
| ProstT5 Heinzinger et al. (2024) | ✓ | Encoder-Decoder | 3B | NAR Gen. Bio. 24 | 0.7683 | GitHub |
| GearNet Zhang et al. (b) | | GNN | 20M | ICLR 23 | 0.5860 | GitHub |
| **Sequence-Function** | | | | | | |
| ProtST Xu et al. (2023) | ✓ | Encoder | 750M | ICML 23 | 0.7176 | GitHub |
| ProLLaMA Lv et al. (2025) | | Decoder | 6.7B | IEEE TAI 25 | 0.5475 | GitHub |
| OntoProtein Zhang et al. (a) | | Encoder | 420M | ICLR 22 | 0.6287 | GitHub |
| **Sequence-Structure-Function** | | | | | | |
| ProCyon Queen et al. (2024) | | Decoder | 11B | Arxiv 24 | 0.1909 | GitHub |
| ESM3 Hayes et al. (2025) | ✓ | Encoder | 1.4B | Science 25 | 0.6483 | GitHub |
| ProTrek Su et al. (2024) | ✓ | Encoder | 650M | Arxiv 24 | 0.7641 | GitHub |

adjusts rank allocation during training Mao et al. (2024). **DoRA** decomposes weights into magnitude and direction for targeted adaptation Zhang et al. (2023). We implement these methods using the PEFT library Mangrulkar et al. (2022).

**Hyper-parameters.**  All models are trained for up to 50 epochs using AdamW with a batch size of 64 and early stopping after 5 patience epochs. Optimal learning rate is selected from {1e-5, 1e-4}.

## 4 EXPERIMENTS

We conduct systematic experiments to answer the following questions:

- **Q1: Supervised Tuning.** How are different supervised downstream tasks correlated, and can a minimal, representative subset of tasks be identified to efficiently benchmark pre-trained models?

- **Q2: Zero-shot Evaluation.** Can zero-shot protocols reliably evaluate protein foundation models?

- **Q3: PEFT Strategies.** Which PEFT methods are more effective for protein tasks?

- **Q4: Scaling.** How does model performance improve with increased model size?

### 4.1 SUPERVISED TUNNING (Q1)

**Task Correlations.**  We evaluate the adapter tuning performance of 12 core models across 28 core tasks, with the complete results provided in the appendix (Table 7) due to space constraints. We analyze task relationships using Spearman correlation and visualize the results in Figure 3, where p-values greater than 0.05 are marked with ✗. Finally, the 28 core tasks are grouped into 11 clusters based on their correlations, and the selected **representative tasks** (marked as ☆).

**Core Model Performance on Representative Tasks.**  Table 3 summarizes the performance of 12 core models on 11 representative tasks. Poorly performing tasks are excluded due to the challenges adapter tuning faces in learning them. Upon analyzing the poorly performing datasets, we observe that the newly implemented 30% sequence identity split introduces significant challenges for model learning. While the stability performance under the original split aligns with SaProt Su et al., the new split proves to be more demanding. Interaction tasks, requiring paired sequence embeddings processed via transformer adapters, remain particularly challenging, underscoring the need for PLMs tailored for interaction prediction, as current models are trained solely on single sequences.

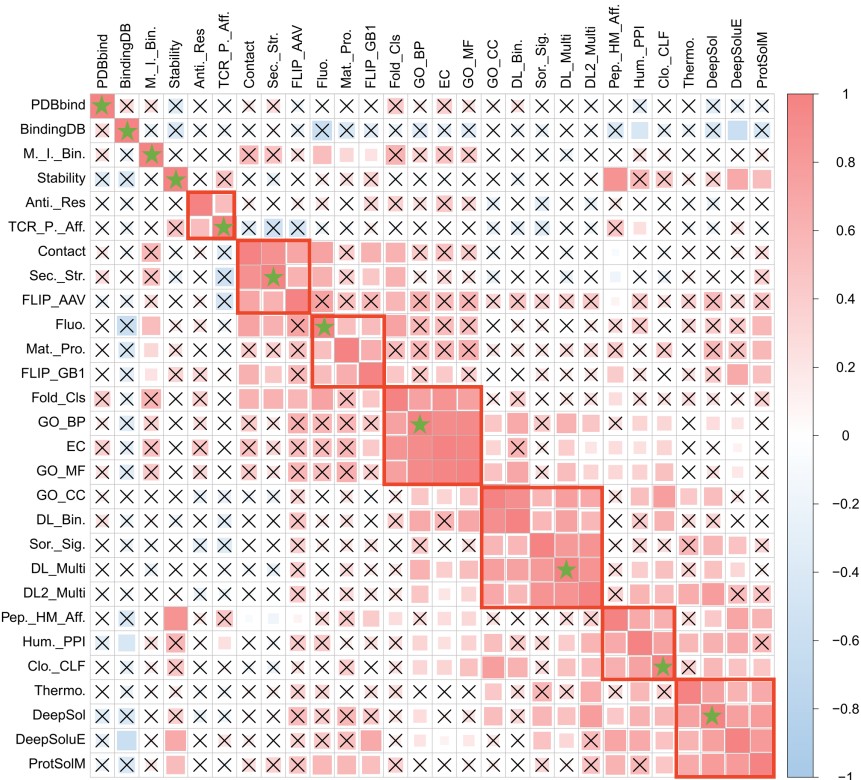

Figure 3: Task relations in supervised tuning.

Table 3: Core model results on representative tasks. **Best** and second-best ones are highlighted.

| | PDBBind | Bind. DB | Stability | Anti.Res. | Mat.Pro. | EC | M. I. Bin. | Sec. Str. | DL2 M. | Clo. CLF | DeepSol | #Win |
|---|---|---|---|---|---|---|---|---|---|---|---|---|
| **Sequence** | | | | | | | | | | | | |
| ESM-2 Lin et al. (2023) | 0.14677 | 0.13692 | 0.32112 | 0.63422 | 0.81189 | 0.73578 | 0.71170 | 0.76375 | 0.76191 | 0.80586 | **0.84494** | – |
| VenusPLM Tan et al. (2025) | 0.16536 | 0.16834 | **0.33907** | 0.64602 | **0.82018** | 0.75194 | 0.70195 | 0.71637 | 0.73814 | 0.83172 | 0.82775 | 50% |
| ESM-C | 0.14692 | 0.20716 | 0.29976 | 0.67257 | 0.81009 | 0.71694 | 0.70195 | 0.76777 | 0.75395 | 0.81033 | 0.84171 | 38% |
| ProtGPT2 Ferruz et al. (2022) | 0.13503 | 0.17169 | 0.14803 | 0.67257 | 0.76757 | 0.69687 | 0.71170 | 0.49371 | 0.70341 | 0.77730 | 0.78883 | 13% |
| PGLM Chen et al. (2024) | 0.16877 | 0.16884 | 0.33127 | 0.67257 | 0.79495 | 0.74659 | 0.74513 | 0.72842 | 0.74772 | **0.83638** | 0.82160 | 50% |
| ProtT5 Elnaggar et al. (2021) | **0.20105** | 0.19730 | 0.18638 | 0.68732 | 0.80072 | 0.76201 | 0.72145 | 0.77978 | 0.72624 | 0.78485 | 0.78741 | 50% |
| DPLM Wang et al. (2024b) | 0.13659 | 0.17408 | 0.29440 | 0.68732 | 0.80144 | 0.75521 | 0.70056 | 0.75695 | 0.75759 | 0.81247 | 0.82841 | 38% |
| **Sequence-Structure** | | | | | | | | | | | | |
| SaProt Su et al. | 0.15549 | 0.16557 | 0.24804 | 0.65782 | 0.81081 | 0.75144 | 0.71031 | **0.82389** | 0.74006 | 0.81206 | 0.84364 | 50% |
| ProstT5 Heinzinger et al. (2024) | 0.18344 | 0.16642 | 0.13032 | **0.69027** | 0.81622 | **0.76829** | 0.72145 | 0.81397 | 0.73190 | 0.79853 | 0.81937 | 63% |
| **Sequence-Function** | | | | | | | | | | | | |
| ProtST Xu et al. (2023) | 0.19514 | 0.18886 | 0.06623 | 0.63422 | 0.69261 | 0.71761 | 0.51532 | 0.68468 | 0.74886 | 0.80714 | 0.81951 | 13% |
| **Sequence-Structure-Function** | | | | | | | | | | | | |
| ESM3 Hayes et al. (2025) | 0.15572 | **0.22519** | 0.15650 | 0.58407 | 0.77514 | 0.64830 | 0.70334 | 0.81264 | 0.65853 | 0.77391 | 0.78106 | 13% |
| ProTrek Su et al. (2024) | 0.17322 | 0.19230 | 0.04924 | 0.59292 | 0.81477 | 0.76408 | **0.80362** | 0.77363 | **0.83944** | 0.82612 | 0.83427 | **75%** |

**Do existing PLMs truly outperform ESM2?** For the remaining 8 representative tasks, we compare each model against ESM2 and calculate the winning rate (#Win), which is defined as the proportion of tasks where a model outperforms ESM2. From the #Win metric, we observe that:

- **Sequence-based PLMs.** All sequence-based PLMs achieve no more than a 50% winning rate against ESM2, indicating that they could not outperform ESM2 on the representative tasks.

- **Decoder-only Model.** The decoder-only model ProtGPT2 performs the worst on these tasks, with a winning rate of only 13% on representative tasks. This suggests that the decoder-only architecture is currently unsuitable for protein understanding.

- **Multimodal PLMs.** Multimodal PLMs achieve the highest winning rates, with ProTrek attaining a 75% winning rate on representative tasks. This success is attributed to the effective semantic alignment of sequence and function information during the pre-training stage.

- **Challenges with Function Data.** ESM3 and ProtST show low winning rates (13%) due to noisy or insufficient function data, emphasizing the need for high-quality, large-scale datasets. For example, ProTrek excels when trained on such cleaned, large-scale annotations.

## 4.2 ZERO-SHOT EVALUATION (Q2)

While ProteinGym has established itself as a valuable zero-shot benchmark for protein function prediction, recent studies (Tsishyn et al., 2025; Gurev et al.; Zhou et al., 2024) have revealed limitations in its sensitivity to model size and performance differentiation. We greatly appreciate ProteinGym's seminal role in standardizing zero-shot mutation evaluation and accelerating progress in protein ML. PFMBench complements ProteinGym by focusing on more supervised learning scenarios—including mutation tasks— that better align with real-world deployment needs, while introducing novel evaluation metrics (e.g., Mutual Information Difference) to assess functional sequence relationships. This complementary approach provides a more comprehensive evaluation framework for protein foundation models.

**Complementary Evaluation Insights.** Zero-shot mutation tests are a common practice, yet Table 4 shows that ProteinGym's zero-shot performance has limited correlation with supervised tuning results, indicating the mutation tasks and predictive tasks probe different facets of model capability. This underscores PFMBench's role as a complementary suite that include both mutation and predictive tasks, thereby providing signals that are orthogonal to mutation assessments. In addition, we introduce a new zero-shot evaluation metric (Mutual Information Difference) to better align the model's ability with sequence-level mutual information.

Table 4: Zero-shot proteingym performance of core models.

| | # Params | Architecture | Input | Loss | ProteinGym | Rank |
|---|---|---|---|---|---|---|
| SaProt Su et al. | 650M | Encoder | Seq-Struct | MLM | 0.45094 | 1 |
| VenusPLM Tan et al. (2025) | 300M | Encoder | Seq | MLM | 0.43952 | 2 |
| ESM-2 Lin et al. (2023) | 650M | Encoder | Seq | MLM | 0.43904 | 3 |
| ESM-C | 600M | Encoder | Seq | MLM | 0.43422 | 4 |
| DPLM Wang et al. (2024b) | 650M | Encoder | Seq | MLM | 0.42922 | 5 |
| ESM3 Hayes et al. (2025) | 1.4B | Encoder | Seq | MLM | 0.41401 | 6 |
| PGLM Chen et al. (2024) | 1B | Encoder-Decoder | Seq | MLM | 0.39750 | 7 |
| ProTrek Su et al. (2024) | 650M | Encoder | Seq | MLM+ Contrast | 0.35919 | 8 |
| ProtGPT2 Ferruz et al. (2022) | 738M | Decoder | Seq | NTP | 0.18962 | 9 |

**UMAP Visualization.** Figure 4 shows UMAP embeddings of ESM2, ProstT5, and ProTrek on Deeploc2_Multi, colored by class labels. ESM2 and ProstT5 exhibit overlapping clusters, while ProTrek, leveraging contrastive alignment, shows distinct boundaries. This highlights the importance of semantic alignment in pretraining for capturing functional relationships.

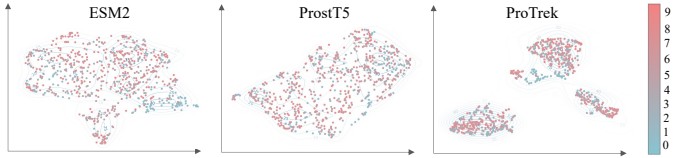

Figure 4: UMAP visualization of ESM2, ProstT5, and ProTrek on Deeploc2_Multi.

**MSA Mutual Information.** We compute the Mutual Information Difference (MID) for sequence-only models relative to ESM2-35M across 100 MSA clusters (see Appendix A.4 for MID definition). MSA centers are randomly sampled from UniRef30 Suzek et al. (2015), with mmseq2 Steinegger & Söding (2017) used for top-10 MSA searches. Figure 5 shows that ProTrek and larger ESM models achieve higher MID, consistent with their downstream performance, suggesting that PLMs effectively clustering local MSA.

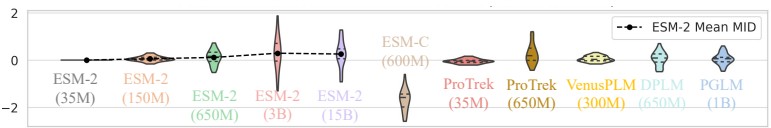

Figure 5: The MID distribution of sequence-only models relative to ESM2-35M.

### 4.3 OPTIMAL EFFICIENT FINE-TUNING AND SCALING (Q3 & Q4)

Table 5 presents the performance of the top-2 models alongside the ESM2 baseline on 11 representative tasks using various efficient fine-tuning methods, including Adapter, Linear Probing, LoRA, AdaLoRA, DoRA, and IA3. For each fine-tuning method, we calculate the winning rate (#WESM) against ESM2. Additionally, across different fine-tuning methods, we compute the winning rate (#WAdap) against the adapter tuning method for each model. We observe that:

- **Adapter Tuning is Sufficiently Effective.** The adapter tuning method consistently outperforms other fine-tuning methods across all models, except for DoRA.
- **ProTrek Consistently Outperforms ESM2.** ProTrek achieves the best performance across all fine-tuning methods, with a winning rate of 75% to 88% against ESM2.

Table 5: Results on 11 representative tasks using various efficient fine-tuning methods. PEFT methods that outperform the Adapter are marked in red; the others are marked in blue.

| | PDBBind | BindingDB | Stability | Anti. Res. | Mat. Prod. | EC | M. I. Bin | Sec. Str. | DL2 Multi | Clo. CLF | DeepSol | #WESM | #WAdap |
|---|---|---|---|---|---|---|---|---|---|---|---|---|---|
| **Adapter** | | | | | | | | | | | | | |
| ESM-2 | 0.14677 | 0.13692 | 0.32112 | 0.63422 | 0.81189 | 0.73578 | 0.71170 | 0.76375 | 0.76191 | 0.80586 | 0.84494 | – | – |
| ProstT5 | 0.18344 | 0.16642 | 0.13032 | 0.69027 | 0.81622 | 0.76829 | 0.72145 | 0.81397 | 0.73190 | 0.79853 | 0.81937 | 63% | – |
| ProTrek | 0.17322 | 0.19230 | 0.04924 | 0.59292 | 0.81477 | 0.76408 | 0.80362 | 0.77363 | 0.83944 | 0.82612 | 0.83427 | 75% | – |
| **Linear Probing** | | | | | | | | | | | | | |
| ESM-2 | 0.21766 | 0.16427 | 0.04649 | 0.64307 | 0.81586 | 0.61163 | 0.71309 | 0.71846 | 0.74472 | 0.78807 | 0.79465 | – | 38% |
| ProstT5 | 0.24874 | 0.16144 | 0.05228 | 0.67552 | 0.80541 | 0.65167 | 0.66992 | 0.79928 | 0.70530 | 0.78722 | 0.77794 | 38% | 0% |
| ProTrek | 0.22595 | 0.25353 | 0.02332 | 0.63717 | 0.81766 | 0.64201 | 0.69777 | 0.73840 | 0.77847 | 0.80099 | 0.80312 | 75% | 25% |
| **LoRA** | | | | | | | | | | | | | |
| ESM-2 | 0.18463 | 0.24559 | 0.32304 | 0.61652 | 0.80865 | 0.67146 | 0.69499 | 0.74305 | 0.76851 | 0.83616 | 0.86160 | – | 38% |
| ProstT5 | 0.19072 | 0.21411 | 0.28204 | 0.66077 | 0.81658 | 0.72779 | 0.64485 | 0.80878 | 0.77875 | 0.82997 | 0.84834 | 63% | 0% |
| ProTrek | 0.24707 | 0.19302 | 0.2776 | 0.67257 | 0.84324 | 0.71139 | 0.74373 | 0.76687 | 0.79566 | 0.83441 | 0.86326 | 88% | 50% |
| **AdaLoRA** | | | | | | | | | | | | | |
| ESM-2 | 0.20398 | 0.23794 | 0.26715 | 0.60767 | 0.80829 | 0.68715 | 0.71448 | 0.7436 | 0.77209 | 0.84171 | 0.85077 | – | 50% |
| ProstT5 | 0.21487 | 0.07897 | 0.17776 | 0.68142 | 0.82883 | 0.71974 | 0.66156 | 0.80755 | 0.75642 | 0.82935 | 0.85272 | 63% | 50% |
| ProTrek | 0.24625 | 0.22491 | 0.15328 | 0.64307 | 0.83640 | 0.7384 | 0.68524 | 0.76651 | 0.80497 | 0.83713 | 0.86152 | 75% | 50% |
| **DoRA** | | | | | | | | | | | | | |
| ESM-2 | 0.18497 | 0.20087 | 0.33022 | 0.63717 | 0.82739 | 0.68786 | 0.72006 | 0.74357 | 0.77774 | 0.84471 | 0.86346 | – | 75% |
| ProstT5 | 0.23039 | 0.10505 | 0.26731 | 0.69912 | 0.80000 | 0.70583 | 0.66574 | 0.80813 | 0.77520 | 0.83052 | 0.85343 | 38% | 50% |
| ProTrek | 0.23648 | 0.07242 | 0.25293 | 0.60177 | 0.83387 | 0.71772 | 0.72006 | 0.76710 | 0.80063 | 0.83988 | 0.86625 | 63% | 75% |
| **IA3** | | | | | | | | | | | | | |
| ESM-2 | 0.18948 | 0.19144 | 0.09641 | 0.60177 | 0.79928 | 0.68549 | 0.63231 | 0.74286 | 0.76447 | 0.82562 | 0.83062 | – | 25% |
| ProstT5 | 0.24188 | 0.12700 | 0.04821 | 0.66962 | 0.82342 | 0.71467 | 0.71309 | 0.81016 | 0.74326 | 0.78942 | 0.80635 | 63% | 25% |
| ProTrek | 0.23836 | 0.10734 | 0.06299 | 0.59292 | 0.79676 | 0.70588 | 0.71031 | 0.76366 | 0.78881 | 0.82911 | 0.83146 | 75% | 25% |

**Are Scaling PLMs Truly Worth It?** In Table 6, we further examine whether increasing model size improves performance on the 11 representative tasks, focusing on the ESM2 series models. We calculate the winning rate (W150M) of each model against ESM2-150M and conclude the following:

- **Scaling Up Works but Comes at a Cost.** The scaling law is effective only when models are scaled up to 15B parameters; otherwise, none of the models outperform ESM2-150M. However, this increase in model size incurs significant costs in both pretraining and inference. Considering the marginal performance gains, the cost of scaling up may not be justified.
- **Pretraining Strategies Matter More.** Instead of scaling up to 15B, a more effective and efficient approach is to optimize the pretraining strategy. For instance, ProTrek-650M outperforms ESM2-15B on 6 out of 8 tasks and achieves a winning rate of 75% against ESM2-150M.

Table 6: Performance of ESM2 under the scaling law. Gray tasks are excluded from the winning rate analysis. Models that outperform the ESM2-150M are marked in red; the others are marked in blue.

| | PDBBind | Bind. DB | Stability | Anti.Res. | Mat.Pro. | EC | M. I. Bin. | Sec. Str. | DL2 M. | Clo. CLF | DeepSol | #W150M |
|---|---|---|---|---|---|---|---|---|---|---|---|---|
| ESM2-35M | 0.09985 | 0.14232 | 0.32337 | 0.67552 | 0.78595 | 0.71675 | 0.71866 | 0.69609 | 0.73219 | 0.79441 | 0.82486 | 13% |
| ESM2-150M | 0.09371 | 0.13142 | 0.33728 | 0.65192 | 0.81946 | 0.73192 | 0.76462 | 0.73430 | 0.74744 | 0.81531 | 0.82825 | – |
| ESM2-650M | 0.14677 | 0.13692 | 0.32112 | 0.63422 | 0.81189 | 0.73578 | 0.71170 | 0.76375 | 0.80586 | 0.84494 | | 50% |
| ESM2-3B | 0.10479 | 0.12724 | 0.31647 | 0.64012 | 0.80036 | 0.73878 | 0.73955 | 0.77111 | 0.77328 | 0.81031 | 0.83007 | 50% |
| ESM2-15B | 0.08427 | 0.12559 | 0.03018 | 0.68142 | 0.81045 | 0.73259 | 0.73259 | 0.77250 | 0.76714 | 0.80210 | 0.85155 | 63% |
| ProTrek-650M | 0.17322 | 0.19230 | 0.04924 | 0.59292 | 0.81477 | 0.76408 | 0.80362 | 0.77363 | 0.83944 | 0.82612 | 0.83427 | 75% |

## 5 CONCLUSION

This work presents a comprehensive benchmark for evaluating protein foundation models (PFMs) across a diverse range of tasks, accompanied by a streamlined evaluation protocol. Starting with 38 tasks and 17 models, we identify 12 core models and 11 representative tasks to enable efficient and meaningful evaluation. Through extensive experiments, we reveal that current PFM research exhibits a high degree of homogeneity and provide in-depth analysis to guide future research directions.

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
