# A  APPENDIX

## A.1  SUPPORTED TASKS

### TASK1: ANNOTATION

**(Definition & Metric)** Annotation tasks aim to predict functional characteristics of proteins. These tasks include predicting the subcellular localization of proteins (Cellular Component), their biochemical activities (Molecular Function), the broader biological processes they participate in (Biological Process) Ashburner et al. (2000), and their classification number according to the chemical reactions they catalyze (Enzyme Commission) Bairoch (2000). F1 Score is the primary metric.

**(Impact)** Accurate annotation facilitates the identification of protein roles in cellular contexts, aiding in the discovery of novel drug targets and the elucidation of disease pathways.

### TASK2: SOLUBILITY

**(Definition & Metric)** Solubility tasks evaluate a protein's ability to remain soluble under physiological conditions, which is a critical factor for successful protein expression and purification. PFMBench includes datasets such as DeepSol Khurana et al. (2018), DeepSoluE Wang & Zou (2023), ProtSolM Tan et al. (2024b), and eSOL Chen et al. (2021). The primary metrics are AUROC for DeepSoluE, DeepSol, and ProtSolM, and Spearman correlation for eSOL.

**(Impact)** Predicting protein solubility is crucial for the successful expression and purification of recombinant proteins, which are essential in drug development and industrial applications. Insoluble proteins can lead to aggregation, reducing biological activity and complicating downstream processes.

### TASK3: LOCALIZATION

**(Definition & Metric)** Localization tasks focus on predicting the specific subcellular compartments where proteins are localized, which is crucial for understanding protein functions and interaction networks. These tasks include DeepLoc Multi Almagro Armenteros et al. (2017), DeepLoc2 Multi Thumuluri et al. (2022), DeepLoc Binary Almagro Armenteros et al. (2017), and Sorting Signal Thumuluri et al. (2022). The evaluation metrics are Accuracy for DeepLoc Multi, F1 Score for DeepLoc2 Multi and Sorting Signal, and AUROC for DeepLoc Binary.

**(Impact)** Accurate localization prediction aids in deciphering protein functions, interactions, and cellular pathways, contributing to our understanding of cellular organization and dynamics.

### TASK4: MUTATION

**(Definition & Metric)** Mutation tasks evaluate the impact of amino acid substitutions on protein properties, which is pivotal in understanding disease mechanisms and guiding protein engineering. PFMBench includes datasets such as PETA_CHS_Sol, PETA_LGK_Sol, PETA_TEM_Sol Tan et al. (2024a), FLIP_AAV, FLIP_GB1 Dallago et al., TAPE_Stability, TAPE_Fluorescence Rao et al. (2019), and $\beta$-lactamase activity Gray et al. (2018). The primary metric is Spearman correlation for all datasets.

**(Impact)** Understanding the effects of mutations on protein function and stability is vital for elucidating disease mechanisms and guiding therapeutic interventions.

### TASK5: INTERACTION

**(Definition & Metric)** Protein-protein and protein-ligand interactions are fundamental to cellular processes and drug discovery. These tasks include datasets such as Human-PPI Pan et al. (2010), Yeast-PPI Guo et al. (2008), PPI affinity Moal & Fernández-Recio (2012), PDBbind Liu et al. (2017), BindingDB Liu et al. (2007), Metal Ion Binding Hu et al. (2022b), Peptide HLA MHC Affinity Wu et al. (2023), and TCR PMHC Affinity Koyama et al. (2023). The evaluation metrics include AUROC for Human-PPI, Yeast-PPI, Peptide HLA MHC Affinity, and TCR PMHC Affinity; Spearman correlation for PPI affinity, PDBbind, and BindingDB; and Accuracy for Metal Ion Binding.

**(Impact)** Accurate interaction prediction is crucial for understanding cellular signaling pathways, protein complexes, and drug-target interactions, facilitating drug discovery and development.

TASK6: STRUCTURE

**(Definition & Metric)** Structure tasks focus on predicting the structural properties of proteins based on their sequences, which is essential for understanding their function and stability. These tasks include Contact prediction Yang et al. (2020), Fold classification Lo Conte et al. (2000), and Secondary structure prediction Klausen et al. (2019). The evaluation metrics are Top L/5 for Contact prediction, and Accuracy for both Fold classification and Secondary structure prediction.

**(Impact)** Accurate structure prediction enables the understanding of protein mechanisms, the design of novel proteins, and the development of structure-based therapeutics.

TASK7: PRODUCTION

**(Definition & Metric)** Production tasks involve predicting properties that influence protein expression and manufacturing, which are critical for biotechnological applications. Datasets include Optimal pH Gado et al. (2023), DeepET_Topt Li et al. (2022b), Cloning CLF, Material Production Wang et al. (2014), Enzyme Catalytic Efficiency Li et al. (2022a), Antibiotic Resistance Hu et al. (2022b), and Thermostability Jarzab et al. (2020). The evaluation metrics include Spearman correlation for Optimal pH, Enzyme Catalytic Efficiency, and DeepET_Topt; AUROC for Cloning CLF and Thermostability; and Accuracy for Material Production and Antibiotic Resistance.

**(Impact)** Predicting factors that influence expression levels, stability, and yield can optimize production processes, reducing costs and improving scalability.

TASK8: ZERO-SHOT

**(Definition & Metric)** Zero-shot tasks evaluate models' generalization abilities to unseen data without additional training. PFMBench incorporates the ProteinGym dataset Notin et al. (2023), which assesses the robustness and adaptability of models in predicting mutation effects across diverse proteins. Spearman correlation is the primary metric.

**(Impact)** Zero-shot learning is crucial for evaluating models' generalization capabilities, reflecting real-world scenarios where labeled data is scarce or unavailable.

# B THE USE OF LARGE LANGUAGE MODELS (LLMS)

Large Language Models (LLMs) were employed as assistive tools in the preparation of this work. In particular, we used GPT-5 to support minor edits to academic writing, including drafting and refining sections. All scientific claims, methodological contributions, and experimental results were entirely conceived, implemented, and validated by the authors. The authors retain full responsibility for the content of this paper.

## B.1 MORE RESULTS

Table 7 summarizes core model performance across 28 tasks using 6-layer transformer adapters. Sequence-only models performed similarly to ESM2, with no model significantly exceeding the baseline. ProTrek, with contrastive pretraining, achieved the best performance, though potential label leakage from overlapping functional annotation data remains a concern for function-aware models.

The detailed model rankings across different tasks are shown in Fig. 6, with tasks grouped by category. Different models excel at different types of tasks, such as ProTrek for annotation, ESM2 for solubility, and PGLM for interaction. The zero-shot results do not correlate with the supervised tuning results.

Table 7: Adapter tuning performance of core models on core tasks.

| Model | ESM-2 | VenusPLM | ESM-C | ProtGPT2 | PGLM | ProtT5 | DPLM | SaProt | ProstT5 | ProtST | ESM3 | ProTrek |
|---|---|---|---|---|---|---|---|---|---|---|---|---|
| PDBbind | 0.14677 | 0.16536 | 0.14692 | 0.13503 | 0.16877 | 0.20105 | 0.13659 | 0.15549 | 0.18344 | 0.19514 | 0.15572 | 0.17322 |
| BindingDB | 0.13692 | 0.16834 | 0.20716 | 0.17169 | 0.16884 | 0.19730 | 0.17408 | 0.16557 | 0.16642 | 0.18886 | 0.22519 | 0.19230 |
| M. I. Bin. | 0.71170 | 0.70195 | 0.70195 | 0.71170 | 0.74513 | 0.72145 | 0.70056 | 0.76323 | 0.72145 | 0.51532 | 0.70334 | 0.80362 |
| Stability | 0.32112 | 0.33907 | 0.29976 | 0.14803 | 0.33127 | 0.18638 | 0.29440 | 0.24804 | 0.13032 | 0.06623 | 0.15650 | 0.04924 |
| Anti. Res | 0.63422 | 0.64602 | 0.67257 | 0.68437 | 0.67257 | 0.68732 | 0.68732 | 0.65782 | 0.69027 | 0.63422 | 0.58407 | 0.59292 |
| TCR P. Aff. | 0.93190 | 0.93784 | 0.93378 | 0.94002 | 0.94542 | 0.93983 | 0.92470 | 0.89967 | 0.93078 | 0.91649 | 0.86510 | 0.90497 |
| Contact | 0.71755 | 0.58946 | 0.72026 | 0.07141 | 0.63453 | 0.79012 | 0.71687 | 0.83507 | 0.82642 | 0.52120 | 0.76616 | 0.73618 |
| Sec. Str. | 0.76375 | 0.71637 | 0.76777 | 0.49371 | 0.72842 | 0.77978 | 0.75695 | 0.82389 | 0.81397 | 0.68468 | 0.81264 | 0.77363 |
| FLIP_AAV | 0.93848 | 0.92354 | 0.93936 | 0.33732 | 0.87888 | 0.93825 | 0.94491 | 0.94822 | 0.93977 | 0.92250 | 0.92514 | 0.93999 |
| Fluo. | 0.68116 | 0.66353 | 0.65043 | 0.61042 | 0.66926 | 0.67662 | 0.67930 | 0.69642 | 0.68020 | 0.56488 | 0.66469 | 0.66987 |
| Mat. Pro. | 0.81189 | 0.82018 | 0.81009 | 0.76757 | 0.79495 | 0.80072 | 0.80144 | 0.81081 | 0.81622 | 0.69261 | 0.77514 | 0.81477 |
| FLIP_GB1 | 0.95306 | 0.94869 | 0.95772 | 0.86281 | 0.91945 | 0.95217 | 0.92162 | 0.95133 | 0.95408 | 0.82742 | 0.88144 | 0.94049 |
| Fold Cls | 0.77546 | 0.75460 | 0.73067 | 0.64724 | 0.77055 | 0.82761 | 0.79448 | 0.80552 | 0.82761 | 0.72577 | 0.72515 | 0.80613 |
| GO BP | 0.54411 | 0.54212 | 0.51338 | 0.48536 | 0.52669 | 0.55179 | 0.55989 | 0.56237 | 0.53352 | 0.41313 | 0.61936 |  |
| EC | 0.73578 | 0.75194 | 0.71694 | 0.69687 | 0.74659 | 0.76201 | 0.75521 | 0.75144 | 0.76829 | 0.71761 | 0.64830 | 0.76408 |
| GO MF | 0.64062 | 0.66136 | 0.60517 | 0.58921 | 0.64860 | 0.65762 | 0.66604 | 0.65340 | 0.68076 | 0.62999 | 0.54672 | 0.71195 |
| GO CC | 0.61448 | 0.62054 | 0.61501 | 0.58498 | 0.61593 | 0.60873 | 0.62185 | 0.62047 | 0.60785 | 0.63078 | 0.52218 | 0.70202 |
| DL Bin. | 0.90619 | 0.91855 | 0.90482 | 0.90117 | 0.91495 | 0.90736 | 0.93305 | 0.92042 | 0.91657 | 0.94016 | 0.90032 | 0.94336 |
| Sor. Sig. | 0.87027 | 0.80974 | 0.85391 | 0.77861 | 0.81180 | 0.79012 | 0.83804 | 0.81408 | 0.82789 | 0.87278 | 0.79688 | 0.86161 |
| DL Multi | 0.75899 | 0.73502 | 0.76165 | 0.68442 | 0.72437 | 0.69907 | 0.78029 | 0.72042 | 0.73236 | 0.76698 | 0.62051 | 0.80826 |
| DL2 Multi | 0.76191 | 0.73814 | 0.75395 | 0.70341 | 0.74772 | 0.72624 | 0.75759 | 0.74006 | 0.73190 | 0.74886 | 0.65853 | 0.83944 |
| Pep. H/M Aff. | 0.96347 | 0.96616 | 0.96046 | 0.90498 | 0.96638 | 0.95677 | 0.96310 | 0.94768 | 0.95392 | 0.94323 | 0.93000 | 0.94650 |
| Hum. PPI | 0.85095 | 0.82147 | 0.83961 | 0.79784 | 0.87692 | 0.81359 | 0.85760 | 0.85100 | 0.79113 | 0.80034 | 0.72483 | 0.84690 |
| Clo. CLF | 0.80586 | 0.83172 | 0.81033 | 0.77730 | 0.83638 | 0.78485 | 0.81247 | 0.81206 | 0.79853 | 0.80714 | 0.77391 | 0.82612 |
| Thermo. | 0.95036 | 0.91701 | 0.94953 | 0.91401 | 0.94224 | 0.92826 | 0.93949 | 0.96930 | 0.91747 | 0.94393 | 0.87837 | 0.93172 |
| DeepSol | 0.84494 | 0.82775 | 0.84171 | 0.78883 | 0.82160 | 0.78741 | 0.82841 | 0.84364 | 0.81937 | 0.81951 | 0.78106 | 0.83427 |
| DeepSoluE | 0.77630 | 0.74926 | 0.76009 | 0.68645 | 0.75549 | 0.72004 | 0.74118 | 0.75492 | 0.74905 | 0.72849 | 0.67909 | 0.73090 |
| ProtSolM | 0.85874 | 0.84107 | 0.85452 | 0.79735 | 0.84894 | 0.80456 | 0.84847 | 0.85718 | 0.84728 | 0.79923 | 0.80773 | 0.83168 |

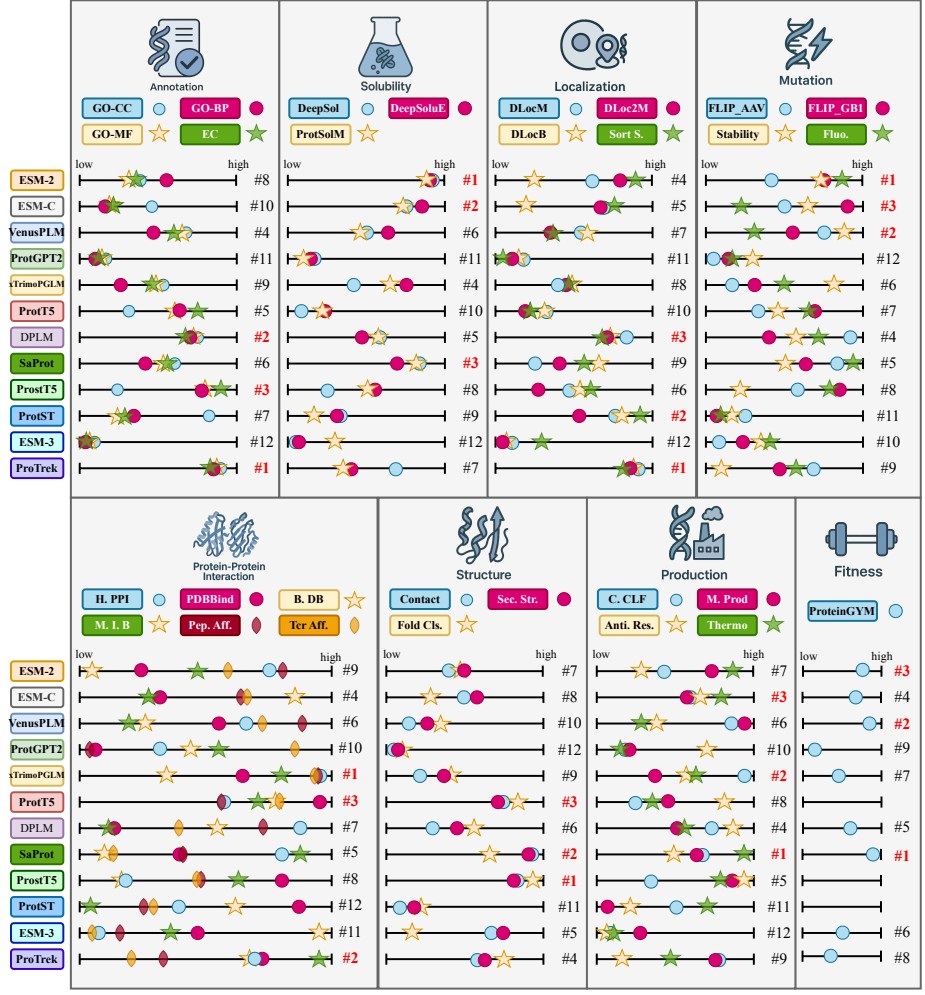

Figure 6: Model rank on tasks.

## B.2 MUTUAL INFORMATION

**Mutual Information Difference Metric.** For a set of MSA sequences $\{x^{(0)}, x^{(1)}, x^{(2)}, x^{(3)}, \cdots\}$, we compute the mutual information (MI) Tschannen et al.; Poole et al. (2019); McAllester & Stratos (2020) between the target sequence $x^{(0)}$ and a query sequence $x^{(i)}$. When the two MSA sequences differ in length, the mutual information is computed only over their aligned and overlapping regions. The mutual information is defined as:

$$I(x^{(i)}; x^{(0)}) = \sum_{k \in \mathcal{I}} \log \frac{p(x_k^{(i)} \mid x_{/k}^{(0)})}{p(x_k^{(i)})},$$

where $\mathcal{I}$ represents the set of mask indices, $p(x_k^{(i)} \mid x_{/k}^{(0)})$ denotes the conditional probability of the $k$-th token in $x^{(i)}$ predicted by a PLM given the context of $x_{/k}^{(0)}$, $x_{/k}^{(0)}$ indicates that the $k$-th residue is masked, and $p(x_k^{(i)})$ refers to the marginal probability when the input is fully masked. We use a PLM to estimate $p_\theta(x_k^{(i)} \mid x_{/k}^{(0)})$ and compute the MI difference between different PLMs. Taking ESM2-35M as the base model $\theta_0$, the MI difference for a new model $\theta_1$ is defined as:

$$I(x^{(i)}; x^{(0)}, \theta_1) - I(x^{(i)}; x^{(0)}, \theta_0) = \sum_{k \in \mathcal{I}} \log \frac{p_{\theta_1}(x_k^{(i)} \mid x_{/k}^{(0)})}{p_{\theta_0}(x_k^{(i)} \mid x_{/k}^{(0)})}.$$

## B.3 CRITERION FOR SELECTING CORE MODELS

We selected EC classification as our representative benchmark based on a systematic evaluation of tasks with performance bias below 0.1%. As shown in Table 8, EC classification emerges as the optimal choice for three compelling reasons:

**Computational Efficiency:** With only 13,090 training samples, EC classification has the smallest dataset among all low-bias tasks, making it computationally accessible and efficient for evaluation.

**Scientific Validity:** EC classification is well-established in the AI-for-biology literature and has been extensively validated in prior work (Fan et al., 2022; Zhang et al., b; Hu et al., 2024b;a; Hua et al., 2024), ensuring robust baseline comparisons and meaningful scientific interpretation.

**Performance Stability:** The task exhibits extremely low performance bias (0.09%), making it a stable and reliable benchmark for fair model comparison.

Table 8: Tasks with performance bias below 0.1%.

| Task | #Train | Bias(%) |
|------|--------|---------|
| Human-PPI | 30133 | 0.00 |
| Secondary Structure | 67007 | 0.00 |
| Pept.HLA/MHC Aff | 57357 | 0.00 |
| Material Production | 22196 | 0.00 |
| TCR PMHC Affinity | 19264 | 0.00 |
| EC | 13090 | 0.09 |

Table 9: Inclusive 85% threshold across tasks.

| Model | Yeast-PPI | FoldPrediction | Localization | TCR–pMHC Affinity | Antibody Resistance |
|-------|-----------|----------------|--------------|-------------------|---------------------|
| ESM2 | 0.6307 | 0.7902 | 0.7683 | 0.8995 | 0.6254 |
| ProGen2 | 0.52343 | 0.4724 | 0.5686 | 0.5232 | 0.5487 |
| GearNet | 0.51921 | 0.0200 | 0.5068 | 0.5240 | 0.4425 |
| ProLLaMA | 0.53137 | 0.4178 | 0.6059 | 0.7169 | 0.5428 |
| ProCyon | 0.57145 | 0.0900 | 0.5054 | 0.5000 | 0.3717 |

**Inclusive Threshold:** Our 85% performance threshold (relative to ESM-2) was strategically designed to balance inclusivity with rigor. As demonstrated in Table 9, this threshold successfully captures all candidate models across diverse tasks while maintaining meaningful performance differentiation.