# OpenReview forum: "PFMBench: Protein Foundation Model Benchmark"
_ICLR.cc/2026/Conference — Submitted to ICLR 2026_

### Official Review · Reviewer_yCpT · 2025-11-01

**Soundness:** 3
**Presentation:** 2
**Contribution:** 2
**Rating:** 6
**Confidence:** 3

**Summary:**

This work introduces a new benchmark for predicting various protein-related attributes using protein foundational models. The tasks considered are broad and include 38 tasks, such as annotation, mutation, structure, and zero-shot learning. This work adds to an expanding collection of benchmarking tools for protein foundational models.
Editorial Issues:
- There were multiple references to the appendix, but the appendix was not included in the submission.
- Fig 3 lacks a proper legend; including the significance of X and stars would be helpful. Currently, it is only described in the text.

**Strengths:**

- The tasks considered span a wide range. This is beneficial for future work to compare.
- The benchmark comparison was done using multiple PEFT methods—mainly Adapter and DoRA. This allows examining the hidden potential of PFMs and not strictly limiting it by its training objectives.
- The inclusion of multi-modal models such as ESM3, is helpful, as these are recent models.
- Assuming the source code for this benchmarking will be made open, it would be a valuable resource, and due to its claimed modular design (which cannot be evaluated currently), it has the potential to significantly aid future work.

**Weaknesses:**

- While the datasets/tasks benchmark works are valuable for advancing AI and are critical for developing the next generation of AI models, the work does not provide novel innovation or understanding of AI.
- A wide range of tasks was considered, but the motivation for including such a broad number of tasks is not explained.
- There is no measure of the quality of the tasks. For example, gene annotations could be noisy, and additional efforts might be necessary to ensure the quality of these datasets.
- The paper relies heavily on tabular comparisons and win rates (#Win) without deeper qualitative analysis.

**Questions:**

- The analysis or discussion of the results from UMAP is missing. Only three selected UMAP plots are provided, but more than 30 tasks and 17 models were considered. What about the results of other models and tasks ?

---

### Official Review · Reviewer_SFzi · 2025-11-01

**Soundness:** 2
**Presentation:** 2
**Contribution:** 2
**Rating:** 4
**Confidence:** 4

**Summary:**

This work proposes PFMBench, a comprehensive benchmark for Protein Foundation Models (PFMs). The authors evaluate 17 state-of-the-art PFMs across 38 tasks spanning 8 key areas of protein science, and develop a protocol that filters the extensive testbed down to 11 representative tasks, 12 core models, 2 recommended baselines, and 2 recommended PEFT methods.

**Strengths:**

1. This work is the first to systematically incorporate and evaluate PFMs that leverage sequence, structure, and functional data modalities. The focus on a "streamlined protocol" is the paper's main contribution.
2. The authors move beyond simple leaderboards to provide actionable insights. The comparison between ProTrek, ESM3, and ProtST, for example, generates a valuable hypothesis about why multimodal models succeed or fail.
3. The paper provides a large-scale, controlled comparison for PFMs. The experimental design is sound. The hierarchical filtering methodology is well-justified.

**Weaknesses:**

1. The most significant omission is the complete absence of generative tasks. PFMs are not just used for "understanding" (discriminative tasks) but increasingly for "creation" (generative tasks) like inverse folding, de novo backbone design, and sequence generation. By focusing only on 38 discriminative tasks, the benchmark overlooks a massive and critical component of PFM capabilities.
2. The protein structures are from AFDB, which may introduce biases into the evaluation.
3. Fair comparison between protein foundation models seems impossible, considering the different pretraining data used in various models. Since the test dataset might have an overlap with the pretraining dataset.
4. This submission is titled "Protein Foundation Model Benchmark", but it missed some of the related works. For example, many works that belong to the "Seq-Struct" category are not included.
5. As a benchmark, no codes are provided for review. Thus, I am not sure about the quality of the benchmark code.

**Questions:**

1. Following on the weakness above: What was the rationale for excluding generative tasks, such as protein design, from the 8 "key areas of protein science" surveyed in this benchmark?
2. For the conclusion that ProTrek's success stems from "effective semantic alignment" while ESM3's failure is due to "noisy or insufficient function data". However, other multimodal PFMs use similar alignment techniques. Could the authors elaborate on what makes ProTrek's strategy uniquely effective compared to its multimodal peers? Is it purely a matter of data quality, or are there unexamined architectural or objective function differences that are more critical?
3. The authors cite "Appendix A.5" (Line 256) for detailed reasons, but this appendix appears to be missing. Furthermore, the appendix numbering seems inconsistent with the main text (Line 420 refers to A.4). Could the authors please provide the missing justification, correct the numbering, and include supporting data to demonstrate that EC classification performance is a reliable proxy for a model's general capabilities across the other 27 core tasks?
4. Is the data copyright clarified?

**Details Of Ethics Concerns:**

None.

---

### Official Review · Reviewer_1BAp · 2025-11-04

**Soundness:** 2
**Presentation:** 2
**Contribution:** 2
**Rating:** 2
**Confidence:** 4

**Summary:**

This paper proposes PFMBench, a benchmark for evaluating protein foundation models (PFMs) across extensive tasks and models (including sequence-only, sequence-structure, sequence-function, and multimodal variants). The authors aim to address gaps in existing benchmarks (e.g., TAPE, PEER, Venus) by including more multimodal models, simplifying evaluation via a "streamlined protocol" (11 representative tasks, 12 core models, Adapter/DoRA tuning), and analyzing task correlations, zero-shot performance, and parameter-efficient fine-tuning effectiveness. Experiments find that ProTrek (multimodal) outperforms baselines like ESM2, and scaling model size yields limited gains compared to optimizing pretraining strategies.

**Strengths:**

Task and model coverage breadth: PFMBench aggregates 38 tasks (more than prior benchmarks like Venus’s 22) and includes 17 models (vs. Venus’s 3), covering multimodal PFMs (e.g., ESM3, ProTrek) often omitted in existing benchmarks. This breadth could, in theory, offer a more comprehensive view of PFM performance.

**Weaknesses:**

1. Benchmark design is incremental, not transformative. PFMBench merely expands the number of tasks/models from prior work (e.g., TAPE→PEER→Venus→PFMBench) without introducing new evaluation paradigms, metrics, or task designs. For example:
    - The 38 tasks are all existing (e.g., Enzyme Commission classification, ProteinGym zero-shot) with no novel tasks that test understudied PFM capabilities (e.g., functional cross-species generalization, de novo design validation).
    - The "streamlined protocol" (filtering 11 representative tasks via correlation) is a trivial application of Spearman correlation, a standard practice in benchmarking (e.g., GLUE for NLP), and provides no new framework for task selection.
    - No new evaluation metrics: PFMBench relies entirely on existing metrics (F1, AUROC, Spearman, Top L/5) and adds only "Mutual Information Difference (MID)"—a minor variant of mutual information (McAllester & Stratos, 2020) with no validation that it better reflects PFM quality than standard metrics.

2. Key conclusions are trivial or already known. For example, "multimodal models outperform sequence-only models": this has been established by ESM3. PFMBench merely confirms this with more tasks, adding no new insight; "Scaling model size yields limited gains": ESM2’s scaling analysis already showed diminishing returns beyond 650M parameters. PFMBench’s ESM2 series results are redundant.

**Questions:**

n/a

---

### Official Review · Reviewer_SvfA · 2025-11-04

**Soundness:** 2
**Presentation:** 2
**Contribution:** 2
**Rating:** 2
**Confidence:** 4

**Summary:**

This paper introduces PFMBench, a comprehensive benchmark designed to evaluate Protein Foundation Models (PFMs). The authors argue that existing benchmarks are insufficient as they often cover a limited number of tasks. PFMBench addresses this gap by curating 38 tasks across 8 categories and evaluating 17 state-of-the-art models. Through task-correlation analysis, the authors filter the initial set down to 28 "core tasks" (by removing high-bias tasks) and subsequently to 11 "representative tasks". This paper evaluates various types of models, including sequence-only models, sequence-structure models, sequence-function models, and sequence-structure-function models. Through extensive experiments, the authors compare the pros and cons of different model types on downstream tasks, the effectiveness of various fine-tuning techniques, and the impact of scaling on model performance.

**Strengths:**

1. The benchmark makes comprehensive evaluation that incorporating 38 tasks and 17 models, including many large-scale PFMs (>500M parameters) that were absent in some benchmarks like TAPE.
2. The proposed PFMBench includes and analyzes multimodal PFMs (e.g., sequence-structure, sequence-function), which is an area that not well studied in previous benchmarks.
3. The benchmark includes and compares six different PEFT methods (e.g., Adapter, LoRA, DoRA), offering insights into which methods are most effective.

**Weaknesses:**

1. The evaluation is restricted to "understanding" tasks and completely ignores "generation" tasks. Generation is a critical capability of modern foundation models and is explicitly supported by several models discussed (e.g., ESM3, DPLM, Progen).
2. The selection of the "core models" relies on performance on a single task: Enzyme Commission (EC) classification. This is a questionable methodology. Although the authors justify this choice in Appendix B.3, the reasoning appears weak. For instance, the authors claim the EC task has low performance bias (0.9%), but it is unclear why a task with even lower (or 0%) bias was not chosen. Regardless, using a single, specific task for model selection is highly likely to introduce selection bias, which could compromise the validity of the subsequent experimental conclusions.
3. Some of the paper's conclusions are either trivial or have been well-established by prior work: the findings in lines 370-375 that decoder-only models are unsuitable for understanding tasks [1] and that multinomial pLMs outperform sequence-only pLMs [2][3] are already known. Such superficial conclusions offer little new insight to the reader.
4. The conclusion in line 376 that attributing the lower performance of ESM3 and ProtST to "noisy or insufficient function data" lacks reliable evidence. There may be other plausible factors. For example, could this be related to the source and quality of the sequence data, not just the function data? Since ESM3's training also includes structure data, will this have an impact? Also, will the training objectives of ESM3 and ProtST influence the result? The authors do not provide sufficient evidence to support this specific claim.
5. In Section 4.3, the authors do not justify why only ProtT5, ProTrek, and ESM2 were selected to validate the impact of fine-tuning techniques. It is unclear whether the conclusions drawn from just these three models can be generalized to the broader set of foundation models.
6. The paper's presentation suffers from a lack of essential experimental details, such as a clear description of the inputs and outputs for each task. Furthermore, the definition and purpose of the novel evaluation metric (Mutual Information Difference) should be presented in the main body of the paper, instead of in the appendix.
7. The paper's conclusions are largely based on observing and summarizing experimental results, but lacks sufficient insights into the reasons for these results. Providing more in-depth insights into why specific models or fine-tuning techniques perform better would be highly beneficial, as it would offer valuable guidance to readers for developing improved models.

[1] BERT: Pre-training of Deep Bidirectional Transformers for Language Understanding

[2] SaProt: Protein Language Modeling with Structure-aware Vocabulary

[3] ProTrek: Navigating the Protein Universe through Tri-Modal Contrastive Learning

**Questions:**

See above weaknesses.

---

### Meta-Review · Area_Chair_Hcke · 2026-01-06

**Summary:**

The reviewers have concerns about the lack of generation tasks, lack of clarity on what each task is and why it was included, and the lack of new tasks or insights about the determinants of performance.

**Reviewer Concerns:**

**No generative tasks**

This is an important frontier (arguably the most important) for protein foundation models.

**Many tasks are not clear, and there is no code.**

For many tasks, it is not clear what the inputs and outputs are!

**This is mainly a compilation of existing tasks with no new tasks or insights.**

**Reviewer Scores:**

No rebuttal, so no changes.

---

### Decision · Program_Chairs · 2026-01-26

Reject